# The Impact of “Home Office” Work on Physical Activity and Sedentary Behavior during the COVID-19 Pandemic: A Systematic Review

**DOI:** 10.3390/ijerph191912344

**Published:** 2022-09-28

**Authors:** Patrick Wilms, Jan Schröder, Rüdiger Reer, Lorenz Scheit

**Affiliations:** 1Department I–Internal Medicine, Bundeswehr Hospital Hamburg, Lesserstr. 180, 22049 Hamburg, Germany; 2Department of Sports Medicine, Faculty for Psychology and Human Movement Science, Institute for Human Movement Science, University of Hamburg, Turmweg 2, 20148 Hamburg, Germany

**Keywords:** home office, work-from-home, physical activity, sedentary behavior, lockdown, COVID-19 pandemic

## Abstract

In 2020, as part of the COVID-19 pandemic, governments around the world enacted a wide variety of regulations and laws to contain the incidence of infection. One of these measures was the relocation of work to the home office. The objective of this review was to analyze the influence of the home office in correlation with regulations on sedentary and activity behavior. A search was conducted on various electronic databases from November 2019 to January 2022, using the search terms physical activity (PA), COVID-19, and working from home. The primary outcomes were changes in PA and sedentary behavior (SB). Secondary outcomes included pain, mood, and parenting stress. The risk of bias was assessed using the (NHLBI) Quality Assessment Tool. For the review, 21 articles met the inclusion criteria (total n = 1268). There was a significant increase in SB (+16%) and a decrease in PA (−17%), Light PA (−26%), and moderate to vigorous PA (−20%). There was also an increase in pain and parenting stress and a decrease in well-being. Due to our significant results, programs that promote movement should be created. Future studies should explore how an increase of PA and a reduction of SB in the home office could be achieved.

## 1. Introduction

In the context of the COVID-19 pandemic, a wide variety of regulations and laws were enacted by governments around the world in order to contain the incidence of infection [1]. These measures caused wide-ranging restrictions in the everyday life of the population, e.g., closures of schools, stores, and offices, as well as restrictions of the personal freedom of movement of individuals [2]. In order to be able to work anyway, many employees worked from home. For Germany, an increase in home office work to 30% in 2020 (2019: 4%) was verified [3]. Olsen et al. described that the home office causes an increase in the duration of sedentary time (SB) even outside pandemics, while changes in physical activity (PA) tend to play a subordinate role [4]. However, in a multinational survey of more than 13,000 individuals distributed across all continents, moderate-intensity (MVPA) and vigorous-intensity (VPA) physical activities were found to have decreased by 41% and 42%, respectively, as a result of pandemic restrictions [1]. These data are based on a survey using the Nordic Physical Activity Questionnaire in its short form (NPAQ-SF), in which physical activity in the moderately strenuous (MVPA) and very strenuous (VPA) segments—and here, separately for work-associated or leisure-associated activities for the time before and during lockdown—was recorded. A distinction between aerobic endurance and strengthening activity as recommended by the WHO or as in the European Health Interview Survey-Physical Activity Questionnaire (EHIS-PAQ) was not made here. Methodological approaches may explain differences in the magnitude of activity declines in different studies. Accordingly, data vary with respect to PA decreases for different activity segments from −24% for moderate PA and −22% for VPA to −41% for MVPA [5] and −42% for VPA [1].

The multinational global declines in moderate and vigorous activity described by Wilke et al. (2021) were increasingly related to times associated with work rather than leisure activities [1]. Here, a particular relevance for the type of work is anticipated: home office or teleworking, or “working from home”, could be a significant influencing factor. Similarly, the “stay-at-home-order” shows comparable effects on physical activity.

In this regard, the COVID-19 pandemic has a unique feature. Due to the dynamic development of the pandemic, many were unable to set up the conditions recommended for the home office, such as a closed office or an ergonomic work area [6].

Additional difficulties due to government regulations became apparent. Parents had to provide childcare; social contacts had to be restricted; certain leisure activities were no longer possible; activities that were taken for granted, such as shopping, were made more difficult by entry restrictions [2].

The objective of this review, in contrast to existing reviews, was not only to classify pandemic-related changes in activity behavior in general, but to highlight the influence of the home office on changes in sedentary and activity behavior in correlation with national restrictions.

## 2. Materials and Methods

This review was registered on the Prospero review database in accordance with the PRISMA 2020 checklist in a timely manner on 18 February 2022, prior to the start of data analysis [7] (Registration Number: CRD42022311435).

### 2.1. Search Strategy and Inclusion Criteria

A systematic literature search was conducted in the scientific databases “PubMed”, “Web of Science”, and “Livivo”. Thereby, the search was limited to the period from 1 January 2020 to 8 February 2022. This revealed 1268 results. The search strategy was based on the population, intervention, controls, outcomes (PICO) scheme [8] and can be found in Table 1. The articles were reviewed with respect to the criteria “statement on physical activity”, “statement on changes in physical activity due to changes in working conditions”, and “statements on associated changes in biological parameters”. The population was narrowed down to male and female subjects in a position of employment over the age of 18 years. All articles not dealing with working adults or dealing with special patient populations were excluded. The articles collected were reviewed in steps initially, and the systemic review software “Rayyan” (http://www.rayyan.ai, accessed on 27 May 2022) was used. Here, it is possible to distinguish between “Included”, “Excluded”, and “Maybe”. Articles rated “Maybe” were reviewed by a second reviewer. 

Following the recommendations of the PRISMA checklist, the articles were examined step by step (Figure 1). All duplicates were removed, followed by exclusion by title and all identified meta-analyses and reviews. For the remaining original articles, abstracts were screened. Articles that did not meet the inclusion criteria were excluded. Finally, a full-text review was performed.

From the included study data, demographic (gender, age, and origin) and psychological or anthropometric indices (well-being, BMI) of the documented samples were extracted, but most importantly, changes in PA or SB due to altered working conditions were extracted. The results are listed in Appendix A.

The periods of the studies included were September 2019 to July 2021. The “stringency level” was used to account for the different developments of the pandemic and governmental actions taken.

### 2.2. Risk of Bias

The 2013 National Heart, Lung, and Blood Institute (NHLBI) Quality Assessment Tool for Observational Cohort and Cross-Sectional Studies (https://www.nhlbi.nih.gov/health-topics/study-quality-assessment-tools, accessed on 27 May 2022) was used for quality assurance (Appendix B) [9] (Table 2). This tool was developed by the NHLBI to check the internal validity of studies. By answering 14 questions, the reviewer identifies a potential risk of bias, as can be seen in Appendix B.

With the help of the determined quality score, a classification of the study quality according to the NHLBI “Quality Rating” (Good, Fair, or Poor) was performed. A maximum of 14 points could be achieved.

### 2.3. Stringency Level

In order to capture the influences of governmental restrictive measures during the lockdown phases in the different nations, the “Stringency Level” of the “Oxford COVID-19 Government Response Tracker” was chosen as a tool for qualitative description [2]. To calculate the “stringency level”, indices were created that were calculated from government measures (e.g., school closures), the handling of the virus (e.g., testing and vaccination strategies), and state aid for, as well as burdens on the health care system (e.g., emergency investments in the health care system or the failure of lucrative interventions for the care of COVID-19 patients). Using a total of 23 criteria, a “stringency level” between 0 and 100 was thus calculated. A stringency level of 0 corresponds to no restrictions on the individual, while a stringency level of 100 corresponds to a total lockdown.

### 2.4. Content Data Analysis

For transparent comparability across many different recording types of PA and SB (e.g., in min/d, min/week, MET metabolic equivalent of task (MET), or the Baecke Physical Activity Index (BPAI)), data are presented in terms of their percent change if possible and, if possible, also with published effect sizes (ESs)—usually Cohen’s d. The accepted conventions for a small, medium, or strong effect with, respectively, |d| epted c|d| ep0.5 or |d0.8|, were applied to the classification of ESs. If percent change data or effect sizes of these values were not available, the authors of the original articles were contacted and the data were supplemented where possible.

To compare the identified studies as a whole with other papers, we calculated a weighted n-adjusted mean for each PA type and SB. For this purpose, we included the percentage decline related to the number of subjects to evaluate the different study sizes (n = 11 [10] to n = 2466 [11]).

**Table 2 ijerph-19-12344-t002:** Risk of Bias.

Criterion (Additional 1) Study No.	1. Question Stated?	2. Population Specified?	3. Participation ≥ 50%?	4. Inclusion and Exclusion Criteria Uniformly?	5. Sample Size Tested?	6. Exposure Measured Prior to the Outcome?	7. Timeframe Sufficient?	8. Did the Study Examine Different Levels of Exposure?	9. Exposure Measures Clearly Defined?	10. Exposure Assessed More Than Once over Time?	11. Outcome Measures Clearly Defined?	12. Outcome Assessors Blinded?	13. Drop Out ≤ 20%?	14. Confounding Variables Measured and Adjusted Statistically?	Sum Score
Aegerter et al. (2021) [12]	Y	Y	N	Y	N	Y	Y	Y	Y	Y	Y	O	O	N	9
Aladro-Gonzalvo et al. (2021) [13]	Y	Y	N	Y	N	Y	Y	N	Y	N	Y	O	O	N	7
Argus et al. (2021) [14]	Y	Y	Y	Y	N	Y	Y	N	Y	N	Y	O	O	N	8
Barkley et al. (2020) [15]	Y	N	Y	Y	N	Y	Y	Y	Y	N	Y	O	Y	Y	9
Brusaca et al. (2021) [10]	Y	Y	N	Y	N	N	Y	N	Y	Y	Y	O	N	N	7
Fukushima et al. (2021) [16]	Y	N	Y	Y	N	N	Y	N	Y	Y	Y	O	O	N	7
Füzéki et al. (2021) [17]	Y	Y	Y	Y	N	Y	Y	Y	Y	Y	Y	O	O	Y	11
Füzéki et al. (2021) [18]	Y	Y	Y	Y	N	Y	Y	Y	Y	Y	Y	O	Y	Y	12
Füzéki et al. (2021) [19]	Y	Y	Y	Y	N	Y	Y	Y	Y	Y	Y	O	Y	Y	12
Howe et al. (2021) [20]	Y	Y	Y	Y	N	Y	Y	N	Y	N	Y	O	O	N	8
Huner et al. (2021) [21]	Y	Y	Y	Y	N	Y	Y	Y	Y	N	Y	O	O	N	9
Katewongsa et al. (2021) [22]	Y	Y	Y	Y	N	Y	Y	Y	Y	N	Y	O	O	N	9
Koohsari et al. (2021) [11]	Y	Y	Y	Y	N	Y	Y	Y	Y	N	Y	O	Y	Y	11
Limbers et al. (2020) [23]	Y	Y	Y	Y	N	Y	Y	Y	Y	N	Y	O	O	Y	10
Lipert et al. (2021) [24]	Y	N	Y	Y	N	Y	Y	Y	Y	N	Y	O	O	N	8
Rapisarda et al. (2021) [25]	Y	Y	Y	Y	N	Y	Y	Y	Y	Y	Y	Y	O	Y	12
Rees-Punia et al. (2021) [26]	Y	Y	N	Y	N	Y	Y	Y	Y	N	Y	O	Y	Y	10
Schoofs et al. (2022) [27]	Y	Y	Y	Y	Y	Y	Y	Y	Y	N	Y	O	O	Y	11
Schuch et al. (2021) [28]	Y	Y	Y	Y	N	N	Y	Y	Y	N	Y	O	O	Y	9
Xiao et al. (2021) [29]	Y	Y	Y	Y	N	Y	Y	Y	Y	N	Y	O	O	Y	10
Yoshimoto et al. (2021) [30]	Y	Y	N	Y	Y	Y	Y	Y	Y	N	Y	O	O	N	9

Legend: Y = yes; N = no; O = other.

## 3. Results

### 3.1. Search Strategy and Identified Sources

A total of 1268 results were found. Of these, 1264 were in the databases “pubmed” (409 results), “Web of scince” (784 results), and “Livio” (71 results) (Table 1). Four results were determined manually. Three inclusion criteria were defined:Statement on physical activity;Statement on the change in physical activity due to the COVID-19 pandemic (e.g., due to home office, higher workload, changed working conditions);Statement on body measurements (e.g., body weight/BMI).

Articles that did not deal with the physical activity of workers were excluded. Excluded also were articles that dealt with specific patterns of diseases.

### 3.2. Methodological Study Quality (Risk of Bias)

The median sum score in the risk-of-bias assessment of the final included studies was 9. Of the 21 included papers, nine papers were classified as “methodologically high quality” (“good”, 10–12 points); the remaining 12 papers were all “methodologically acceptable” (“fair”, 7–9 points); none were “methodologically inferior” (“poor”, <7) (Table 2).

In accordance with the characteristics of online cross-sectional surveys, the criteria most frequently not met were, on the one hand, sample size estimation (item 5) and, on the other hand, the collection of characteristic values more frequently than once during the observation period (item 10).

During the selection of studies, Schuch et al. [28] did not show a strict differentiation between employees (n = 677) and students (n = 43). Due to the small proportion of students (1.6%) in the total sample with otherwise applicable inclusion criteria, we decided to co-consider this study. The resulting systematic error was acceptable in our view, due to the additional gain in information.

Limbers et al. [23] examined only mothers working in home offices in their study. As this was a selective sample, we assessed the results in the context of other studies to avoid a systematic error.

### 3.3. Content Analysis

Two of the included studies had methodological peculiarities that had to be taken into consideration for the data analysis. One Brazilian study [10] collected accelerometer data, while all other studies used questionnaires and memory protocols as a data basis. The work of Rapisarda et al. [25] was the only one that was a true longitudinal study, i.e., no retrospective survey related to the time before and during the restrictions. The list of included studies can be found in Appendix A.

### 3.4. Change in Behavior

#### 3.4.1. Change in Sedentary Behavior

Our research showed an average increase in SB of +16%. Eight of the nine studies showed an increase in SB between +6% [15,22] and +67% [25] (Table 3). Regional specifics could be identified for SB, which—at least in part—can be explained by the specifics of the national restrictions. For Europe, only the study by Rapisarda et al. was available, which described an increase in SB of +67% [25] in Italy at a stringency level of 64.35 (require closing all but essential).

In the USA, Barkley et al. showed an increase in three subgroups of +6% (staff), +10% (administrators), and +15% (faculty) [14] at a stringency level of 72.69 (require closing all but essential), while Rees-Punia et al. showed in their study an increase of +8% [26] for home office employees at a stringency level of 68.89 (require closing some sectors).

For the Asian region, here Japan, Fukuschima et al. and Koohsari et al. showed an increase in SB of +49% [16] at a stringency level of 25.93 (recommend closing) and +10% [11] at a stringency level of 25 (no closing). In their Thai study, Katewongsa et al. described an increase of +6% [22] at a stringency level of 0 (no closing).

For South America, data were available for Ecuador, with an increase in SB of +20% [13] from Aladro-Gonzalvo et al. and from Brazil from Schuch et al. [28]. Schuch et al. showed an increase of +42% [28] with a stringency level of 74.54 (require closing all but essential), while Brusaca et al. [10] reported the opposite trend. They were the only ones to show a reduction in SB of −3% [10] at a stringency level of 74.54 (require closing all but essential).

As a special feature, it should be mentioned that two studies observed a group of students as a comparison group to regular employees. Alardo-Gonzalvoe et al. described an increase in sitting time of +20% for employees and +26.4% for students [13]. Barkley et al. divided employees into three subgroups (faculty +15.2%, staff +6.3%, administrators +9.9%) and students into two subgroups (undergrad +19.2%, graduate +18.1%) [15]. The peculiarity of the study by Barkley et al. [15] is that the university ordered all employees, as well as students to work from home on November 3, 2020. All classes were then taught in digital formats.

#### 3.4.2. Change in Physical Activity

For Europe, Argus et al. reported a decline in PA for Estonia of −5% [14] at a stringency level of 75 (require closing all but essential). Schoffs et al. reported −10% [27] for the Netherlands at a stringency level of 78.7 (require closing some sectors), Lipert et al. [24] for Poland (“stringency level”: 81.48; require closing some sectors), and Rapisarda et al. [25] for Italy (“stringency level”: 64.35; require closing all but essential), each with −35%.

For the USA, only data from the study by Xiao et al. (−15%) [29] with a stringency level of 72.69 were available. In Japan, Koohsari et al. reported a decrease of −11% [11] with a stringency level of 25 (no closing), and from Australia, Hunter et al. reported a decrease of −20% [21] with a stringency level of 60.19 (recommend closing).

For non-sub-differentiated (LPA/MVPA/VPA) PA—as a globalized surrogate score—a decrease in the sample-size-adjusted mean of −16% was observed (Table 4), which correlated significantly with the respective “stringency level” (r = −0.19, *p* = <0.001) (Figure 2), with two studies showing no change in physical activity [12,15].

#### 3.4.3. Change in Light Physical Activity

Due to different methods and definitions in the original studies, changes in light physical activity (Light PA) were examined either as Light PA, or as “walking time”, or also as PA associated with commuting to work, meaning transport-related PA (TRPA). The extent of Light PA decreased by a sample-size-adjusted mean of −26%, with the greatest decrease (−47%) occurring at the lowest stringency level (25.93) in Japan [16]. For walking, a decrease of −14% was determined. For TRPA, the decrease was −23%.

There are no directly comparable data sets for the light activities, because of different methods. Five studies reported a decrease in Light PA of −4% (Rees-Punia et al. [26], the USA, “stringency level” 68.89, require closing some sectors), −11% (Brusaca et al. [10], Brazil, “stringency level” 74.54, require closing all but essential), −30% (Lipert et al. [10], Poland, “stringency level” 81.48, require closing some sectors), −31% (Rapisarda et al. [25], Italy, “stringency level” 64.35, require closing all but essential), and −7% (Fukuschima et al. [16], Japan, “stringency level” 25.93, no closing). Brusaca et al. also reported accelerometer data [10]. Rapisarda et al. also presented the only longitudinal study here [25] (Table 5).

Four studies reported data on walking. Füzeki et al. showed a decrease of −11% [17,18] (Italy/Germany, “stringency level” 85.19/76.85, require closing all but essential) and −17% [19] (Germany, “stringency level” 76.85, require closing some sectors) in three different studies. Rapisarda et al. reported a decrease of −31% for Italy (“stringency level” 64.35, require closing all but essential) [25] (Table 6).

In a study sequence at the time of the first (March 2020 to June 2020) and second (November 2020 to April 2021) lockdown in Germany, the factor “teleworking” was explicitly asked [17,18]. Since the study populations of just under 1000 participants were very similar in their demographic characteristics, a follow-up concept can be assumed for these studies. Only during the second lockdown was it possible to show a significant influence of the home office on work-associated physical activity (TRPA). Data for TRPA were available for four studies and three by Füzeki et al. (−13.6% [18], −20% [19], and −52% [17]), and Koohsari et al. with −9.4% [11] from Japan (“stringency level”, no closing) (Table 6). During the first lockdown in Germany, a significant effect for home office influence on the decline in leisure-time-associated physical activity (LTPA) was found: the LTPA decline was generally about −16%; for non-home office working participants, the decline was −20%; for home office working participants, it was only −11%, while there was no significant effect of home office working status on working-time-associated physical activity (TRPA) [18]. In the second lockdown in Germany, the effects of home office working status on the generally still observed declines in leisure-time-associated physical activity (LTPA) were no longer observed; instead, in the second lockdown in Germany, a significant home office influence on working-time-associated physical activity (TRPA) was observed, whereby the reduction without the home office was about −8% and with the home office about −35% [19].

The three studies by Füzeki et al. [17,18,19] indicated changes in days of muscle-strengthening activities per week (DMSA). They described a decrease of −6% on average.

#### 3.4.4. Change in Moderate to Vigorous Physical Activity

Two studies from Japan were available for the Asian region: Koohsari et al. reported a decrease in moderate PA of −9% and in high PA of −10% [11] (“stringency level” 25, no closing), while Fukushima et al. reported a decrease in MVPA of −40% [16] (“stringency level” 25.93, recommend closing).

Data from two Brazilian studies were available from South America. Brusaca et al. reported a decrease in MVPA of −42% [10] (“stringency level” 74.54, require closing all but essential) and Schuch et al. of −64% [28] (“stringency level” 74.54, require closing some sectors).

For North America, Howe et al. reported a decrease in MVPA in Canada of −44% (“stringency level” 74.54, require closing all but essential) and for the USA −45% (“stringency level” 72.69, require closing all but essential) [20]. Limbers et al. reported a decrease of −50% for moderate PA and −10% for high PA (USA, “stringency level” 72.69, require closing all but essential). Rees-Punia et al. also provided data for the USA and described a decrease of −4% [26] (“stringency level” 68.89, require closing some sectors).

Data from Europe were available from Lipert et al. and Rapisarda et al. Both studies separated moderate and high PA. Lipert et al. reported no change in high PA for their Polish population, while they described a 1% increase in moderate PA [24] (“stringency level” 81.48, require closing some sectors). Rapisarda et al. showed a 4% increase in moderate PA in their Italian study, while high PA decreased by −13% (“stringency level” 64.35, require closing all but essential) [25].

Two studies reported an increase in moderate physical activity of 1% and 4% [24,25]; the others reported a decrease of up to −64% [28]. In the sample-size-adjusted mean, there was a −20% decrease in MVPA, but with very high variability (+1% to −64%) (Table 7). The decrease in MVPA correlated significantly with the respective stringency level (r = −0.11, *p* = <0.001) (Figure 3).

### 3.5. Secondary Outcomes

A deterioration in physical health (pain) was described in three of the included studies [14,29,30]. Seven out of 21 studies showed that there was a higher decline in physical activity of female persons compared to the decline of physical activity of male subjects [13,17,20,22,23,27,29]. The study by Rapisarda et al. showed a +7% increase in BMI [25].

Four studies reported a deterioration in the corresponding parameters of “Well-Being” (−24%) [17], work-comfort/work-ability (ES −0.26/−20%) [14,21], or “Psychological Quality of Life” (−72%) [23]. In contrast, one study described an improvement in work–life balance [12].

## 4. Discussion

The objective of this work was to determine the effects of COVID-19-associated restrictions and, in particular, their severity (operationalized by the “Stringency Level”) with special consideration of home office effects on PA and SB. The main results were that an increase in sitting time (SB) and a decrease in all dimensions of PA could be observed. As far as the authors are aware, this is the first review to systematically address the effects of the home office factor on physical activity and sedentary behavior associated with the COVID-19 pandemic.

It is well known that prolonged sitting and physical inactivity pose a health risk for orthopedic and internal diseases [31,32,33]. Some of the studies shown compared employees in presence at work, home office employees, and the unemployed and found a stronger increase in SB in the home office than in unemployment or presence at work during the lockdown [26,28]. The studies showed a large range of their SB results. One explanation could be the different observation periods and different study designs.

The authors who reported changes in SB uniformly interpreted their results as a direct consequence of changes due to the home office or the “stay-at-home order”. Rees-Punia et al. emphasized that an increase in SB, in this case by 1.5 h per day, simultaneously meant a decrease in LPA and MVPA and that this represents an unfavorable development [26]. No Information on effect sizes could be derived from the studies shown here, so we limited ourselves to a qualitative description.

Nine out of eleven studies showed a decrease in PA of more than −5% (Figure 2) (Table 4). In their study population, Schoofs et al. [27] found that there was a −10% decrease in PA in the general population. This is less than the mean value of −16% that we determined.

The lack of change in PA was interpreted differently by the authors. It was postulated that the comparatively good weather and reduced working hours were a reason for the lack of change [12,19]. Another explanation is the pre-existing lack of movement [15,34]. Individuals who already showed below-average PA before the COVID-19-related lockdown showed no significant change in their activity level as a result of the lockdown [35].

Causes of very large decreases in PA were parent stress and long observation periods in the study protocol [24,25]. By comparison, another study by McCarthy et al. [34] showed a continuous decrease in PA over 6 months of lockdown (January to June 2020) in the general population in the U.K. At the same time, this could not be confirmed in a longitudinal study in Greece by Bourdas et al. [36], also described outside our identified studies. Here, PA showed an increase in the course of the lockdown (observed period of 6 weeks) after an initial sharp drop and almost reached the pre-lockdown level again at the end of the lockdown [36]. A correlation in the stringency level for the time periods stated in the studies could not be found (McCarthy et al. [34] for Great Britain 5.56/Bourdas et al. [36] for Greece 19.44). In our view, there is a need for further research to understand the reasons for maintaining altered physical activity and to counteract this with targeted measures.

All other studies were consistent in the interpretation of their results and held the lack of social interaction, the lack of commuting, and the home office per se responsible for the decline in PA. The additional effect of “parenting stress” was only reported in one study [24]. Again, no effect size measures could be derived from the study.

The particularity of the different operationalization and definitions of Light PA made comparability difficult. For example, there was no explanation in the studies for the explicit decrease in Light PA. It is also unclear from the studies whether and how TRPA was included. Nevertheless, this could be an important point to explain the decline.

Two studies showed a correlation between the decrease in Light PA and the increase in SB [16,26]. Unlike in the SB, Light PA did not seem to decline as much in the home office as it did among unemployed people [26]. However, additional studies are necessary to make a valid statement. 

The decline in TRPA and walking is interpreted by the authors as a direct consequence of the decline in PA in the context of the home office [11,17,18,19,25]. From this, it can be deduced that commuting to work accounted for an important proportion of the subjects’ total physical activity. However, this was not explicitly investigated in the studies and would have been validated.

For the decrease in DMSA [17,18,19], it must be noted that even before the pandemic (mean = 1.8), only a few had reached the amount of DMSA recommended by the WHO (≥2) [33].

Eleven of the 14 studies reported a decrease in MVPA (Figure 3). Again, there was a significant correlation with the stringency level (r = −0.11). A special feature of this section was that four of the studies separated moderate and high physical activity, while the other ten summarized it as MVPA.

Howe et al. described the closure of public parks, gyms, and sports center as a result of government measures as a reduction in the autonomy of the population to choose their own type of PA [20]. This barrier, in combination with the compulsory wearing of medical masks and the recommended restriction of social contacts, can be assumed to be the cause of the sometimes sharp decline. It is likely that parts of MVPA were work-related. However, the proportion cannot be deduced from the analyzed studies. As already described in the Introduction, we assumed that a significant part of MVPA was what due to leisure behavior. Two of our studies calculated leisure time PA (LTPA) for this purpose and described changes from −15.9% [18] to −30% [17], which was similar to the mean value of −20% (n-adjusted) that we determined. We therefore assumed that the decrease in MVPA had a lower correlation with the home office. Nevertheless, other studies have shown that the decline in MVPA during the lockdown was greater among home office workers than among the unemployed [26,27,28]. Here, further studies would be necessary for a substantiated differentiation.

The relevance of the present work is underlined by a long-term study of the Robert Koch Institute (RKI), which showed that unemployed people have poorer physical and mental health and reduced life expectancy due to lower physical activity [37]. The studies we analyzed showed a clear trend, with an increase in SB and a decrease in all forms of PA in the home office. This change is all the more dramatic because the home office population showed a greater decline than the general population and the unemployed included in some studies. It remains to be said that a transfer of the negative long-term outcome of the unemployed to home office employees would be rather speculative, and further studies are needed for an adequate assessment.

In our secondary outcomes, Yoshimoto et al. [30] showed a clear difference between the home office and in-office work with regard to the development of pain. Thus, 23% of home office workers reported an increase in pain with a reduction in PA compared to 11.4% with in-office work. A similar connection between the development of pain due to changed workplace conditions was shown in Estonia [14]. Here, “workplace comfort” and “workplace ergonomics” were also recorded, which were rated worse in the home office than in in-office work. The studies did not determine what influence the changed workplace ergonomics and workplace comfort had on the development of pain. In a study from Quebec, however, it was recommended as early as 2003 to attach importance to workplace ergonomics in order to avoid negative effects such as pain or poor performance [6,16].

The studies showed that there was a higher decline in the physical activity of female persons compared to the decline of the physical activity of male subjects. Several studies we evaluated showed a significantly greater increase in sitting time, as well as a significantly greater decrease in PA for women compared to men [13,17,20,22,27,29]. This was also shown by McCarthy et al. in their study [34]. There are different approaches as to why women are affected by a greater decline. One reason for this could be “parenting stress”, resulting in a lower PA level and a reduced “Physical and Psychological Quality of Life” [23]. In the U.K., the Office for National Statistics showed that women take on about 77% of care work time [38]. Another possible explanation is that women are more likely to use sporting activities in an organized group setting than men [27]. Restrictions had a stronger effect there. Furthermore, women used online courses four-times more often than men. These results could be used to come up with opportunities to increase women’s PA in their free time [39].

Additionally, the lack of BMI increase can be explained by the short observation period of the cross-sectional studies. In a 5-year observational study of more than 2000 children, it was shown that changes in BMI (+2.5 kg/m^2^) can only be observed over a longer period of time (here, 5 years), even with physical inactivity [40].

The home office or stay-at-home order seems to have an influence on well-being. Faulkner et al. [41] showed, outside the studies we examined, a connection among sitting time, PA, and well-being in New Zealand. High sedentary time was associated with a poor well-being score [41]. The same applied to low physical activity. Only a few of the studies we reviewed provided data on this. Most of them reported a decrease in well-being in the home office on different measurement instruments [21,23]. At the same time, not all studies have been able to show a decrease or correlate this with the home office [12,19]. Nevertheless, it is considered certain that a decrease in PA can increase depression symptoms, or conversely, that increased PA can improve them [11,23]. Whether negative and positive effects of the home office can equalize and whether there is a direct correlation between home office and reduced well-being should be further investigated.

Our work has strengths and limitations. Overall, the studies we extracted showed a low bias risk. Nevertheless, there are potential bias risks, since, with one exception, no longitudinal studies were analyzed. A bias due to non-published studies cannot be ruled out either. Another limiting factor was that, with the exception of one study by Brusaca et al. [10], only self-completed questionnaires were used for data collection. The data from Brusaca et al. reported a −3% reduction in SB [10]. In addition to the small and very selective sample of eleven office workers, however, the special feature of this study was that SB was quantified with the help of an accelerometer, so that direct comparisons with questionnaire data were not advisable, especially because this was not about PA, but about SB, which is much more difficult to differentiate [42]. The self-completed questionnaires data were often linked to memory protocols from the pre-COVID-19 period. However, these data were collected on the basis of very different measurement methods, including the BPAI, the International Physical Activity Questionnaire (IPAQ), the Global Physical Activity Questionnaire (GPAQ), and the European Health Interview Survey-Physical Activity Questionnaire (EHIS-PAQ). Five of the studies reported mild PA, while four others reported walking and TRPA. As neither walking nor TRPA are coherently defined, a direct correlation with mild PA is not possible exactly. In addition, the studies did not provide any information on the work structure (e.g., open-plan office versus individual workstation) of the study participants, so that comparability is also lacking here. A strength of our review is the fact that we used the “Oxford COVID-19 Government Response Tracker” [2] in order to map the national regulations for comparisons of internationally varying restriction conditions. However, cultural, climatic, or geographical characteristics could not be ruled out. 

## 5. Conclusions

The aim of our study was to analyze the relationship between the home office and a decrease in physical activity during the COVID-19 pandemic restrictions. As a main result, we confirmed significant reductions of PA and identified significant increases of SB due to pandemic-related changes of working conditions, which, in turn, were significantly correlated with the nationally varying amount of COVID-19 governmental restrictions. Thus, our findings revealed additional information on the impact of COVID-19-related governmental restrictions on health behavior with a specific view on the role of working from home conditions. Facing the fact that it is well known that sufficient physical activity is the prerequisite for a healthy life, we conclude in line with earlier systematic reviews that a strategy to increase physical activity should be an important task for employees, as well as for employers and the public health system. Counteracting measures could be offered, for example, within the framework of occupational health management promotion as online offers or as organized face-to-face programs in order to reduce negative health effects for individuals, as well as the associated socioeconomic burdens for the national health systems. Future studies should investigate whether the negative changes in SB and PA were reversible after the end of the restriction measures and possible long-term health consequences for employees.

## Figures and Tables

**Figure 1 ijerph-19-12344-f001:**
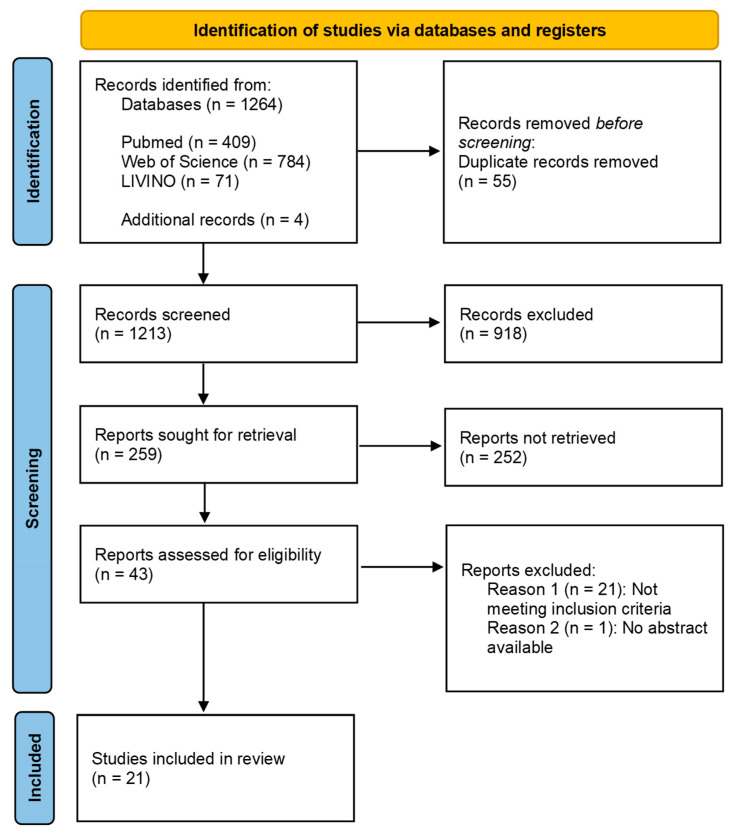
PRISMA 2020 flow diagram [7].

**Figure 2 ijerph-19-12344-f002:**
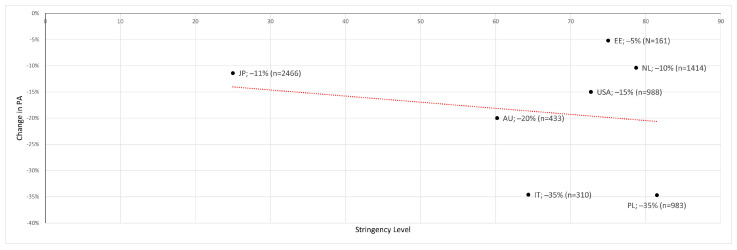
Change in PA to stringency level with trend line (red) [11,14,21,24,25,27,29].

**Figure 3 ijerph-19-12344-f003:**
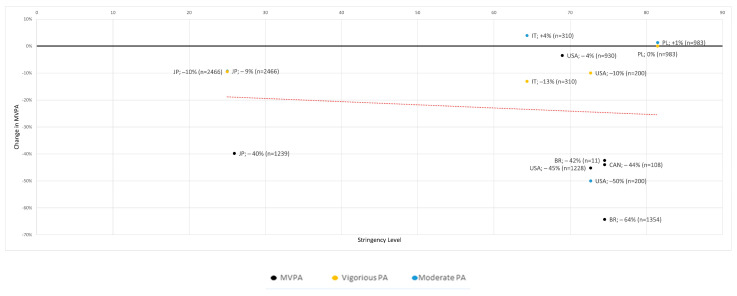
Moderate to vigorous PA to stringency level with trend line (red) [10,11,16,20,23,24,25,26,28].

**Table 1 ijerph-19-12344-t001:** Search strategy and identified sources.

Data Base	Key Words	Results
PubMed	(((physical activity[Title]) OR physical behaviours[Title]) AND COVID-19[Title]) AND ((Working from home) OR (home office) OR (Home Working) OR (Remote Working))	409
Web of Science	(((((TI = (physical activity)) AND TI = (physical behaviours)) AND TI = (COVID-19)) AND ALL = (Working from home)) OR ALL = (home office)) OR ALL = (Home Working) OR (Remote Working)	784
LIVIVO	Title and free text-filters: 1. Abstracts; 2. Articles of the last 2 years; 3. Language: German and English ((TI = (physical activity) AND Working from home) AND TI = (COVID-19)) AND PY = 2020:2022	71
Additional	“hand search in identified articles’ references”	4
Identification summed up (8 February 2022)		1268

**Table 3 ijerph-19-12344-t003:** Change in sedentary behavior.

SB	Participants	Change
Barkley et al. (2020) [15]	10	10%
Brusaca et al. (2021) [10]	11	−3%
Barkley et al. (2020) [15]	28	6%
Alardo-Gonzalvo et al. (2021) [13]	67	20%
Barkley et al. (2020) [15]	176	15%
Rapisarda et al. (2021) [25]	310	67%
Rees-Punia et al. (2021) [26]	930	8%
Fukushima et al. (2021) [16]	1239	49%
Schuch et al. (2021) [28]	1354	42%
Koohsari et al. (2021) [11]	2466	10%
Katewongsa et al. (2021) [22]	6531	6%

**Table 4 ijerph-19-12344-t004:** Change in PA.

PA	Participants	Change
Aegerter et al. (2021) [12]	76	0%
Argus et al. (2021) [14]	161	−5%
Rapisarda et al. (2021) [25]	310	−35%
Hunter et al. (2021) [21]	433	−20%
Lipert et al. (2021) [24]	983	−35%
Xiao et al. (2021) [29]	988	−15%
Schoofs et al. (2022) [27]	1414	−10%
Koohsari et al. (2021) [11]	2466	−11%

**Table 5 ijerph-19-12344-t005:** Change in LPA.

Light-PA	Participants	Change
Brusaca et al. (2021) [10]	11	−11%
Rapisarda et al. (2021) [25]	310	−31%
Rees-Punia et al. (2021) [26]	930	−4%
Füzeki et al. (2021) [18]	979	−16%
Lipert et al. (2021) [24]	983	−30%
Füzeki et al. (2021) [19]	993	−22%
Fukushima et al. (2021) [16]	1239	−47%

**Table 6 ijerph-19-12344-t006:** Change in TRPA/walking.

TRAP/Walking	Participants	Change TRPA/Walking
Rapisarda et al. (2021) [25]	310	n.a./−31%
Füzeki et al. (2021) [18]	979	−13.6%/−11%
Füzeki et al. (2021) [19]	993	−20%/−17%
Füzeki et al. (2021) [17]	1500	−51.9%/−11%
Koohsari et al. (2021) [11]	2466	−9.4/n.a.

**Table 7 ijerph-19-12344-t007:** Change in moderate to vigorous PA.

MVPA	Participants	Change
Brusaca et al. (2021) [10]	11	−42%
Fukushima et al. (2021) [16]	1239	−40%
Howe et al. (CAN) (2021) [20]	108	−44%
Howe et al. (USA) (2021) [20]	1228	−45%
Koohsari et al. (moderate PA) (2021) [11]	2466	−9%
Koohsari et al. (vigorous PA) (2021) [11]	2466	−10%
Limbers et al. (moderate PA) (2020) [23]	200	−50%
Limbers et al. (vigorous PA) (2020) [23]	200	−10%
Lipert et al. (moderate PA) (2021) [24]	983	1%
Lipert et al. (vigorous PA) (2021) [24]	983	0%
Rapisarda et al. (moderate PA) (2021) [25]	310	4%
Rapisarda et al. (vigorous PA) (2021) [25]	310	−13%
Rees-Punia et al. (2021) [26]	930	−4%
Schuch et al. (2021) [28]	1354	−64%

## Data Availability

The data from the current study are presented in the review and are available from the corresponding author upon request.

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
