# Peer review of "The Impact of “Home Office” Work on Physical Activity and Sedentary Behavior during the COVID-19 Pandemic: A Systematic Review"

_ijerph, 2022, doi:10.3390/ijerph191912344_

Round 1

Reviewer 1 Report

The research carried out in preparing this publication was very thorough.  The results show the growing importance of the topic and the wide range of problems that can be encountered when studying the impact of the COVID epidemic.

The empirical research methodology was developed and explained in a way that was understandable to a responsive and broad audience, in line with the aims of the thesis.

The results are well presented because the paper shows important coherence between the introduction, the methodology's objectives, and the discussion.  The conclusions are also understandable. 

The results obtained by this thesis could be helpful for several topics.  The paper is clearly explained with good English and understandable jargon.

Some minor modifications are suggested:

- There is no reference to the first figure in the text.

- In the second table, the text of question 8 is not fully visible.

- In line 144, the reference to the table should be modified.

- The third table is a handy and reasonable summary of the results, but due to its size, it might be more appropriate to include it as an annex.

- An unnecessary hyphen has been added to the word 'infor-mation' in line 468.

- In line 476, there is a redundant hyphen between the word 'so-cio-economic'.

Author Response

Dear Reviewer,

thank you very much for the constructive evaluation of the article and the helpful and valuable comments on correction and improvement. We have edited the text in the author team and send you the new version and hope that you like the corrections.

Reviewer 1

  • - There is no reference to the first figure in the text.

Reference inserted in line 87

  • - In the second table, the text of question 8 is not fully visible.

The table was adjusted, the text is now fully visible

  • - In line 144, the reference to the table should be modified.

Reference was modified (See line 109 + 118)

  • - The third table is a handy and reasonable summary of the results, but due to its size, it might be more appropriate to include it as an annex.

The third table was extracted and attached as Appendix B

  • - An unnecessary hyphen has been added to the word 'infor-mation' in line 468.

Was corrected

  • - In line 476, there is a redundant hyphen between the word 'so-cio-economic'.

Was corrected

Thank you very much

Lorenz Scheit

Reviewer 2 Report

ABSTRACT

Clear.

INTRODUCTION

Interesting. Adequate.

METHODOLOGY

State the exact duration of the literature search.

Move the results of the search (Table 1, Table 3) to the Result section.

Move the eligibility criteria in Table 1 to the text. Do not combine with the search strategies table.

Cite the PRISMA used in the text. PRISMA 2009 was the older version. Used PRISMA 2020.

Please describe the risk of bias by NHLBI in further detail.

Please elaborate on the stringency level.

RESULTS

The description of the risk of bias tool should be in the Method section.

Any overall results for all the included studies with regards to physical activity, light physical activity, vigorous physical activity?

DISCUSSION

How will you apply the results of risk of bias in the discussion?

Line 300-302. Not relevant to the results. Remove.

Line 303-307. Move to limitation.

Line 308-319. Discussion should be focused on overall results. Not on individual studies.

After line 319 onwards, please modify the discussion. It should be focused on overall results. Not on individual studies. And not repeating the information in the Result section.

Overall, the discussion is inadequate.

Author Response

Dear Reviewer,

thank you very much for the constructive evaluation of the article and the helpful and valuable comments on correction and improvement. We have edited the text in the author team and send you the new version and hope that you like the corrections.

Reviewer 2

  • - State the exact duration of the literature search.

The duration of the literature search was from 01.01.2020 to 08.02.2022 (line 74-75)

  • Move the results of the search (Table 1, Table 3) to the Result section.

Table 1 was moved to the Results section (line 158) and Table 3 was added as Appendix B (as recommended by Reviewer 1).

 Move the eligibility criteria in Table 1 to the text. Do not combine with the search strategies table.

The eligibility criteria were formulated as a separate subchapter 3.1 (line 145) and Table 1 was adapted (line 158).

  • Cite the PRISMA used in the text. PRISMA 2009 was the older version. Used PRISMA 2020.

The new  PRIMSA 2020 was cited in the text (line 70) and updated (line 103).

  • Please describe the risk of bias by NHLBI in further detail.

The risk of bias was further descried in line 109 – 111 

 Please elaborate on the stringency level.

The stringency level were elaborate in line 129 – 130 (0 showed no restrictions on the individual, while a stringency level of 100 corresponds to a total lockdown.)

RESULTS

  • The description of the risk of bias tool should be in the Method section.

The description was shifted to the Method section (line 115 – 118).

 Any overall results for all the included studies with regards to physical activity, light physical activity, vigorous physical activity?

The overall results were inserted as table 3 to 7 into the Result section (line 222, 242, 285, 287 and 318).

DISCUSSION

  • How will you apply the results of risk of bias in the discussion?

Risk of bias is considered as low and discussed in line 511 – 514.

  • Line 300-302. Not relevant to the results.

Lines was removed

  • Line 303-307. Move to limitation.

Was moved to limitation (line 515 – 520)

  • Line 308-319. Discussion should be focused on overall results. Not on individual studies.

The discussion was revised where possible and the results of the individual studies were presented as overall results.

  • After line 319 onwards, please modify the discussion. It should be focused on overall results. Not on individual studies. And not repeating the information in the Result section.

The discussion was modified to focus on the overall results. Repetitions from the Results section have been removed.

  • Overall, the discussion is inadequate.

The authors hope that by revising the discussion an adequate presentation has been achieved.

Thank you very much

Lorenz Scheit

Round 2

Reviewer 2 Report

Corrections are made as suggested.